# Database of Shear Experiments on Steel Fiber Reinforced Concrete Beams without Stirrups

**DOI:** 10.3390/ma12060917

**Published:** 2019-03-19

**Authors:** Eva O. L. Lantsoght

**Affiliations:** 1Politécnico, Universidad San Francisco de Quito, Quito 170901, Ecuador; elantsoght@usfq.edu.ec or e.o.l.lantsoght@tudelft.nl; Tel.: +593-2-297-1700 (ext. 1186); 2Concrete Structures, Department of Engineering Structures, Civil Engineering and Geosciences, Delft University of Technology, 2628 CN Delft, The Netherlands

**Keywords:** beams, database, experiments, flexure, shear, steel fiber reinforced concrete

## Abstract

Adding steel fibers to concrete improves the capacity in tension-driven failure modes. An example is the shear capacity in steel fiber reinforced concrete (SFRC) beams with longitudinal reinforcement and without shear reinforcement. Since no mechanical models exist that can fully describe the behavior of SFRC beams without shear reinforcement failing in shear, a number of empirical equations have been suggested in the past. This paper compiles the existing empirical equations and code provisions for the prediction of the shear capacity of SFRC beams failing in shear as well as a database of 488 experiments reported in the literature. The experimental shear capacities from the database are then compared to the prediction equations. This comparison shows a large scatter on the ratio of experimental to predicted values. The practice of defining the tensile strength of SFRC based on different experiments internationally makes the comparison difficult. For design purposes, the code prediction methods based on the Eurocode shear expression provide reasonable results (with coefficients of variation on the ratio tested/predicted shear capacities of 27–29%). None of the currently available methods properly describe the behavior of SFRC beams failing in shear. As such, this work shows the need for studies that address the different shear-carrying mechanisms in SFRC and its crack kinematics.

## 1. Introduction

When steel fibers are added to the concrete mix, the weak tension properties of the concrete may be improved, as the steel fibers can carry this tension. As a result, steel fiber reinforced concrete (SFRC) has superior material and mechanical behavior for all tension-driven material properties and failure modes. An example of a tension-driven failure mode is shear failure [1,2]. Typically, shear-critical elements are provided with shear reinforcement. However, for certain cases, providing shear reinforcement may not be desirable. One example of such an application is reinforced concrete one-way slabs [3], where using shear reinforcement is often not cost-effective. For other cases, heavy shear reinforcement and the resulting reinforcement congestion make casting concrete difficult [4], especially in high performance high strength beams, so that other solutions may be more practical and may lead to a better execution and performance of the structural element. For these cases, dispersing steel fibers in the concrete mix can improve the shear capacity and reduce or eliminate the need for stirrups.

Bernard proposed the use of steel “splinters” to strengthen concrete in tension as early as 1874 [5]. Nevertheless, practical applications of SFRC are still not widespread. The main barrier to application is that building codes, such as the ACI 318-14 [6] and EN 1992-1-1:2004 [7] do not contain provisions for determining the shear capacity of SFRC. The most noteworthy national codes and guidelines with shear provisions for SFRC are the French recommendations [8,9,10], the German guideline [11], and the Italian guide [12].

The currently available shear equations from codes and guidelines, as well as those reported in the literature are summarized in this work. An analysis of the available expressions shows that the majority are empirical equations. Expressions resulting from an analysis of the mechanics of the problem are scarce, with the exceptions of the extensions [13,14] of the modified compression field theory (MCFT) [15] and the dual potential capacity model [16,17]. None of the existing expressions are based on an analysis of the shear-carrying mechanisms in concrete structures [18]: the capacity of the uncracked concrete in the compression zone [19], aggregate interlock [20], dowel action [21], and residual tension across the crack [22]. For SFRC, the contribution of the residual tension across the crack may be negligible, and instead the contribution of the steel fibers bridging the crack should be analyzed [23]. This lack of understanding of the mechanics of the problem forms a more fundamental barrier to the practical application of SFRC. To optimize structural designs, and reduce the material quantities used in a project, as well as their embodied carbon and environmental impact, it is important to develop better models for the shear capacity of SFRC with longitudinal steel reinforcement without stirrups.

Before better models for the shear capacity of SFRC can be evaluated, it is necessary to gather the available experimental data from the literature. This information can be used to analyze the shortcomings of the current equations, and to carry out parameter studies. This paper presents a unique database of 488 experiments. Smaller databases have been reported or discussed in the literature previously [24,25,26,27,28,29,30], but the current effort has resulted in the gathering of a significantly larger number of datapoints. Moreover, the full database is available as a dataset in the public domain for other researchers [31], which is a step forward as well.

## 2. Methods

### 2.1. Overview of Shear Prediction Equations

The currently available expressions to predict the shear capacity of SFRC beams without stirrups are mostly empirical equations. Besides the empirical equations, some methods have been derived that are (partially) based on the mechanics of the problem. Noteworthy here are extensions of the modified compression field theory (MCFT) [15,32] for SFRC, the dual potential capacity model [16,17], and plasticity-based approaches. For the extension of the MCFT to SFRC several approaches have been followed: describing the constitutive equations of cracked SFRC [33,34], assumptions for smeared cracking in SFRC [35], programming the effect of fibers into the VecTor2 software [13,36], panel testing [37], the development of an engineering model [38] for inclusion in the next version of the *fib* model code [39], and the development of a model that considers the rotation of the individual fibers with respect to the crack plane [40] and its closed-form solution [41]. Hwang’s softened truss model with steel fibers [42] falls in the same category as the MCFT for SFRC. Most of these MCFT-based methods require programming and/or the use of finite element models. The dual potential capacity model [16,17] evaluates the capacity of the concrete in the compression zone and the tension capacity of the SFRC in the tension zone. The drawback of this approach is that these assumptions for the mechanics of the behavior do not reflect all shear-carrying mechanisms in SFRC (capacity of compression zone, dowel action, tension capacity of SFRC in tension zone, aggregate interlock, and arching action [18]). Plasticity-based models have been proposed in the past [43,44,45,46]. While the results of these models seem promising, they require further research and validation.

Since the mechanical models that are available in the literature each have drawbacks as pointed out in the previous paragraph, the current codes and guidelines are based on empirical models. Therefore, in this section, an overview of a selection of currently available empirical prediction equations and code equations is given. These prediction models will be used in Section 3 for comparison to the shear capacities obtained from the literature.

Table 1 gives an overview of the shear prediction equations. All symbols used in Table 1 can be found in the list of notations at the end. The expression by Sarveghadi et al. [28] is a simplification of a matrix-based expression resulting from an analysis testing different artificial neural networks. Many expressions describe the steel fiber properties with the fiber factor *F*. The fiber factor [47] is a metric used for defining the properties of the fibers, taking into account the fiber volume fraction *V_f_*, the aspect ratio *l_f_*/*d_f_*, and the bond properties of the fiber *ρ_f_*:(1)$$F=V_{f}\frac{l_{f}}{d_{f}}\rho_{f}$$

The expression of Kwak et al. [48] follows the form of Zsutty’s empirical equation for the shear capacity of reinforced concrete beams [49], with *v_b_* as given in Equation (4). The Greenough and Nehdi expression [50], which is a simplification of an expression resulting from genetic programming, uses a % for *ρ* instead of the actual reinforcement ratio.

Khuntia et al.’s expression [51] is a proposal to include the effect of fibers on the expression for the shear capacity of ACI 318-14 [6]. Similarly, Sharma’s proposal [52] follows the format of the ACI 318-14 code expression, and links the tensile and compressive strength of concrete through the expression by Wright [53]. Mansur et al. [54] also propose an extension of the ACI 318-14 code expression, using *σ_tu_* as recommended by Swamy and Al-Ta’an [55], which uses the fiber length correction factor *η_l_* from Cox [56], the fiber spacing from Swamy et al. [57], and the bond stress *τ* proposed by Swamy and Mangat [58]. Ashour et al. [59] propose two (sets of) equations: the first equation, Equation (17) is a proposal for extension of the ACI 318-14 [6] expressions, whereas Equations (18) and (19) are based on Zsutty’s equation [49]. Arslan’s equations [60] are also based on Zsutty’s equation [49], with the addition of the determination of the height of the compression zone *c* as proposed by Zararis and Papadakis [61]. However, this method for determining *c* ignores the contribution of the fibers on the horizontal and moment equilibrium of the cross-section.

The shear capacity equation from Bažant and Kim [62], derived from fracture mechanics of quasi-brittle materials, was extended to include the contribution of fibers by Imam et al. [63] as well as Yakoub [64] (first set of equations, Equations (25) through (27)). The second set of equations by Yakoub [64], Equations (28) through (32) is a proposal to include the effect of fibers in the shear expressions from the Canadian code CSA A23.3-04 [65], which is based on the MCFT [15].

The next entries in Table 1 are expressions from codes and guidelines. The expressions from the French recommendations [10] separate the concrete contribution to the shear-carrying capacity from the contribution of the fibers. The determination of the contribution of the fibers requires experimental data of the SFRC mix, as shown in Equations (35) through (38). An additional material safety factor *γ_E_* is added so that *γ_cf_γ_E_* = 1.5. The angle of the compression strut *θ* ≥ 30°. The value of *K* in Equation (36) can be approximated as *K* = 1.25, except when *b_w_* and *h* are less than 5*l_f_*, or the value of *K* can be determined from tension tests on the SFRC mix.

The expressions from the German guideline [11] and RILEM [66] are based on the Eurocode EN 1992-1-1:2004 [7] equations, by adding a term to represent the contribution of the steel fibers. The expressions from the *fib* Model Code [39] are based on EN 1992-1-1:2004 [7], but incorporate the effect of the fibers into the original expression. The Italian guide [12] uses the same expressions as the *fib* Model Code [39], and includes a lower bound for the shear capacity *V_min_*. In the German National Annex of the Eurocode 2, *C_Rd,c_* = 0.15, and this value is used in Equation (41) as well. The following factors are used: *γ_c_* = 1.5, $\gamma_{ct}^{f}$ = 1.25, $\alpha_{c}^{f}$ = 0.85 to account for long-term effects, and $k_{F}^{f}$ = 0.5 for shear. For cross-sections subjected to axial loads, the contribution of the steel fibers cannot be taken into account, as more experimental results are necessary to derive suitable expressions [24]. In the Italian guideline [12], the influence of axial loads is considered in the same way as in EN 1992-1-1:2004 [7]. Since this work deals with elements without axial loads, the formulas have been simplified accordingly. The expressions from the German guideline [11], RILEM [66], the *fib* Model Code [39], and the Italian guide [12] are valid for *ρ* ≤ 2%. For the *fib* Model Code expressions, *C_Rd,c_* = 0.18 and *γ_c_* = 1.5. All notations used in Table 1 are explained in the “List of notations”.

### 2.2. Database of Experiments

#### 2.2.1. Development of Database

The database developed for this study contains 488 experiments of SFRC beams with longitudinal tension reinforcement (mild steel only) and without transverse shear reinforcement failing in shear reported in the literature. The consulted references are: Singh and Jain [4], Sahoo and Sharma [67], Shoaib, Lubell, and Bindiganavile [68] (lightweight beams), Manju, Sathya and Sylviya [69], Arslan, Keskin, and Ulusoy [70], Parra-Montesinos et al. [71], Rosenbusch and Teutsch [72], Sahoo, Bhagat, and Reddy [73] (T-beams), Amin and Foster [74], Tahenni et al. [75], Narayanan and Darwish [76], Cucchiara, La Mendola, and Papia [77], Kwak et al. [48], Lim and Oh [78], Dinh, Parra-Montesinos and Wight [79], Lima Araujo et al. [80], Casanova, Rossi, and Schaller [81], Aoude et al. [82], Minelli and Plizzari [83], Kang et al. [84], Casanova and Rossi [85], Lim, Paramasivam, and Lee [44], Mansur, Ong, and Paramasivam [54], Zarrinpour and Chao [86], Noghabai [87], Randl, Mészöly, and Harsányi [88], Ashour, Hasanain, and Wafa [59], Tan, Murugappan, and Paramasivam [89], Pansuk et al. [90], Kim et al. [91], Sharma [52], Narayanan and Darwish [92], Li, Ward, and Hamza [93], Swamy, Jones, and Chiam [94], Cho and Kim [95], Greenough and Nehdi [50], Kang et al. [96], Dupont and Vandewalle [97] with further information in [98], Swamy and Bahia [99], Batson, Jenkins, and Spatney [100], Zhao et al. [101], Jindal [102], Shin, Oh, and Ghosh [103], Imam, Vandewalle, and Mortelmans [104,105], Huang, Zhang, and Guan [106], Kwak, Suh, and Hsu [107], Roberts and Ho [108], Hwang et al. [109], Spinella, Colajanni, and La Mendola [110], Chalioris and Sfiri [111], Cohen and Aoude [112], Aoude and Cohen [113], Qissab and Salman [114], Furlan and de Hanai [115], Dancygier and Savir [116], Krassowska and Kosior-Kazberuk [117], Yoo and Yang [118], Gali and Subramaniam [119], Zamanzadeh, Lourenco, and Barros [120], Shoaib, Lubell, and Bindiganaville [121], Shoaib [122], Bae, Choi, and Choi [123], and Abdul-Zaher et al. [124]. The database does not include the Keskin et al. [125] experiments, since for these specimens carbon fiber reinforced polymer (CFRP) bars were used as longitudinal reinforcement. The experiments by Khan [126] are excluded, as these specimens are subjected to a combination of shear, bending moment, and torsional moment.

Table A1 gives the database developed for this study. The full spreadsheet is available as supplementary file in .xlsx format available in the public domain [31]. The notations used in this database are given in the “List of notations”. For a number of references [42,44,50,52,54,59,67,69,70,71,72,73,75,76,77,78,80,81,83,84,85,88,89,94,96,97,98,99,100,102,103,104,106,107,109,110,111,112,115,116,117,118,119,123,124] information about the geometry of the support and loading plate was missing. These values were then approximated based on figures of the test setup in the original reference. For rollers, the contact surface was assumed to be 10 mm wide. Most specimens are rectangular beams, but the specimens in [73,81,94,99] are T-beams, in [89,90] I-beams, and in [114] non-prismatic beams. Almost all experiments are on simply supported beams in three- or four-point bending, with exception of the two-span beams in [117] and the special setup by [127] for short spans that does not allow for the development of arching action.

In terms of geometry, references [54,69,76] do not report the total length of the beam specimen. Reference [121] only reports the total length for the largest specimens. For the database entries, a similar overhang is used for the smaller specimens. Reference [54] does not report the span length, but the span and total length are estimated from the technical drawings in the original reference. The total length for the beams in [52,89,97,102] was also estimated based on the technical drawings in the paper. A practical value of overhang over the support is assumed for these cases. The results in [103] are inconsistent: the relation between the maximum load in the figures and the shear stress in the reported table is not clear. The cause of this inconsistency seems to be that the authors did not show the length correctly: the sketched span length *l_span_* appears to be the total length *l_tot_*. This correction is included in the database. References [69,81,115] do not report the effective depth. For the database entries, these values are then calculated back from the *a*/*d* ratio, or based on the rebar diameter and a 10 mm cover, as typically used in laboratory conditions on small specimens. Reference [79] reports different values for the effective depth than what can be calculated from the technical drawings. The values from the drawings are used for the database. The ratio *a_v_*/*d* reported in [117] is 2.7. For the database entries, the size of the support plate measured from the technical drawings is used, and the effective depth is calculated assuming a cover of 10 mm. These assumptions result in *a_v_*/*d* = 2.83; the value of *a_v_*/*d* = 2.7 can’t be reverse-engineered based on the available information. Singh and Jain [4] mention that the smallest dimension of the cross-section should be at least three times the length of the longest fiber in the mix. As can be seen in the database, many experiments do not fulfil this requirement. Regardless of their comment, Singh and Jain proceeded to test specimens that do not fulfil this requirement, for ease of comparison to other test results.

The concrete compressive strength in the database is *f_c,cyl_*, the average concrete compressive strength as measured on cylinders. When the compressive strength is reported from cube specimens, the conversion *f_c,cyl_* = 0.85*f_c,cube_* is used. Reference [102] does not give the concrete compressive strength, but uses 3 ksi (21 MPa) in the presented calculation example. Therefore, the value of *f_c,cyl_* is reported as 21 MPa. Reference [119] does not report the concrete compressive strength. Normal strength concrete of *f_c,cyl_* = 30 MPa is assumed. References [50] and [112] used self-consolidating concrete. For references where the maximum aggregate size is not reported [52,74,82,91,109,115,119,120], a standard laboratory mix with *d_a_* = 10 mm is assumed. References [52,86,114,115,124,127] do not report the yield strength of the steel. For these cases *f_y_* = 420 MPa is assumed. For [108], the yield strength at 0.2% strain from the stress-strain diagram is used for the database.

When the tensile strength of the fibers was not given [50,52,71,89,97,98,100,107,108,110,111,115], the value of *f_tenf_* = 1100 MPa was assumed. For recent references, this assumption is reasonable, as this value is common for commercially available fibers. For the experiments by Batson [100] from 1972, it is only known that low-carbon steel was used for the fibers, but the tensile strength of the fibers is not known. The reported tensile strength for fibers by Ashour et al. [59] is smaller than for any other reference. The same value is reported in the paper in MPa and psi units, which seems to exclude a typing error in the reference. Reference [120] used recycled steel fibers. The properties of these fibers were not discussed in this reference, but for the database entries, reference [128] was consulted. Reference [123] does not report on the fiber type and properties. Therefore, standard commercially available hooked fibers were assumed. For the references where the amount of fibers is given as a mass, the fiber volume fraction is calculated by dividing the mass by 7800 kg/m^3^. When the concrete mix contained a combination of fibers [83], the reported fiber properties are weighted averages of the different fibers. Experiment B59 by [99] contained fibers only in the bottom 90 mm of the cross-section.

The results are given in terms of the sectional shear force at failure *V_utot_*, which includes the contribution of the self-weight, as well as in terms of the failure mode. Since this database includes the contribution of the self-weight, the shear at failure from this database may differ from what is reported in the original reference. For small specimens, the effect is small. For lightweight specimens [68,84], the density as reported in the original reference is taken into account to calculate the contribution of the self-weight. When this value was not reported in the original reference [94], a self-weight of 17 kN/m^3^ was assumed. In some references [81], the sectional shear force at failure *V_max_* or the applied load at failure *P_max_* is not included. Where possible [119,123], the load-displacement diagrams are used to read off this value. When this information was not presented, the experiments were not included in the database for lack of vital information. There is a factor 2 difference between the shear stress at failure *v_max_* in [102] and the value I calculated based on the size of the cross-section and the sectional shear at failure *V_max_*. The database contains this calculated value. What [118] reports as the shear force *V_max_* is actually *P_max_*, as one can see when calculating *v_max_*. The following abbreviations are used for the reported failure modes: B (bond failure of longitudinal reinforcement), DT (diagonal tension), NA (the failure mode for the individual experiment is not given in the original reference, but the text mentions that all experiments resulted in shear failure), S (shear failure), SC (shear-compression failure), S-FL (shear-flexure), ST (shear-tension), and Y (yielding of reinforcement).

#### 2.2.2. Parameter Ranges in Database

This section evaluates the distribution of the values of parameters over the database, in terms of range and shape of the distribution. Table 2 gives the ranges of key parameters in the database. These ranges show that the maximum height that has been tested (1220 mm) is relatively small to evaluate the size effect in shear [62,129,130,131,132,133]. The fiber types that occur in the database are: hooked, crimped, straight smooth, mixed (hooked + straight), fibers with a flat end, flat fibers, round fibers, mill-cut fibers, fibers of straight mild steel, brass-coated high strength steel fibers, chopped fibers with butt ends, recycled fibers, and corrugated fibers. The most frequently used fibers in the database are hooked (63% of all gathered experiments), crimped (22% of experiments), and straight smooth (3%).

Figure 1 shows the distribution of a selection of key parameters in the database. In terms of concrete compressive strength, Figure 1a shows that the results in the database are concentrated in the range of normal strength concrete, with some outliers for high and ultra-high strength concrete. For the reinforcement ratio, one can observe in Figure 1b that most specimens have large amounts of longitudinal steel, as typical for shear experiments where extra tension reinforcement is used to avoid a bending moment failure. The experiments are uniformly distributed in the range from 1.5–3.5% reinforcement. The database shows crowding in the range of small effective depths, see Figure 1c. The experiments are normally distributed in terms of shear span to depth ratio, see Figure 1d, with *a*/*d* = 3.5 as the most frequently used shear span. The histogram of the fiber volume fraction *V_f_*, Figure 1e, shows crowding in the range of 0.5–1.5%. This observation is not surprising, as these fractions are practical values: these fractions result in workable mixes, and serve the purpose of partially (not fully) replacing the mild steel reinforcement. Similarly, the observations for the histogram of the fiber factor *F* in Figure 1f reflect practical considerations and workability of SFRC.

## 3. Results

### 3.1. Parameter Studies

First, the raw data from the database are used to analyze the effect of different experimental parameters on the outcome (sectional shear stress at failure as a result of self-weight and applied load). To eliminate the influence of the concrete compressive strength *f_c,cyl_* on the parameter studies, normalized shear stresses are used. There is, however, quite some disagreement in the literature on the effect of the concrete compressive strength on the shear capacity [134]: should we normalize the shear stress with respect to the square or cube root of the concrete cylinder’s compressive strength? Therefore, I analyzed the normalized shear stress to both the square and cube root of the concrete as a function of the concrete compressive strength. Figure 2 shows the relation between the normalized shear stress and the concrete compressive strength *f_c,cyl_*. These results show that the shear stress should be normalized with respect to the square root of *f_c,cyl_*. The influence of different parameters will thus be studied as a function of the shear stress normalized to the square root of *f_c,cyl_*.

Figure 3 gives an overview of the most important parameters and their influence on the shear stress normalized to the square root of *f_c,cyl_*. Figure 3a shows the influence of the reinforcement ratio *ρ*. Larger reinforcement ratios result in larger normalized shear capacities. This observation is expected, since larger reinforcement ratios result in a larger dowel action capacity [21,135,136], and thus a larger shear capacity. Figure 3b shows the influence of the effective depth *d* on the normalized shear stress. In reinforced concrete, the so-called size effect in shear [62,129,130,131,137,138] is known: the shear stress at failure reduces as the effective depth increases. The analysis of the database shows a small size effect. However, very few experiments on specimens with larger depths are available, as shown in Figure 1c. More experiments are necessary to study the size effect in SFRC. Figure 3c shows the influence of the shear span to depth ratio in terms of *a*/*d*. Note that the linear relation plotted on the graph is presented for consistency with the other figures, but does not accurately present the relation between *a*/*d* and the normalized shear strength. These results show that, just as for reinforced concrete beams, the shear capacity for specimens with *a*/*d* ≤ 2.5 increases for a decrease in *a*/*d*. The development of a compressive strut or arch between the point of application of the load and the support increases the shear capacity through the shear-carrying mechanism of arching action [139,140,141]. This influence can also be expressed as a function of the clear shear span to depth ratio *a_v_*/*d* and the generalized expression *M*/*Vd*. Since almost all experiments in the database are three- or four-point bending tests, the difference between *a*/*d* and *M*/*Vd* lies only in the contribution of the self-weight to *M* and *V*. For small specimens, this effect is negligible. For the current database therefore, the difference between the influence of *a*/*d* and *M*/*Vd* is negligible [142]. The parameter *a_v_*/*d* has a slightly larger influence on the normalized shear stress than *a*/*d*. This observation can be explained by the geometries used for deep beams in the database.

Figure 3d shows the relation between the normalized shear capacity and the fiber volume fraction *V_f_*. The normalized shear stress increases as the fiber volume fraction increases. The reason for this observation is the tension carried by the fibers across the crack. Figure 3e shows the relation between the fiber factor *F* and the normalized shear stress. Comparing Figure 3d,e shows that using the fiber factor *F* is an improvement as compared to using only the fiber volume fraction *V_f_*: less scatter is observed. Other properties of the fibers that were studied [142] were the aspect ratio *l_f_*/*d_f_* and the fiber tensile strength *f_tenf_*. The influence of the aspect ratio is similar to the influence of the fiber factor *F*, with the difference that the scatter on the plot with the fiber factor is smaller than for the plot with the aspect ratio. Small increases in the normalized shear strength were found for increases in the fiber tensile strength *f_tenf_*. Since the fibers typically do not reach their tensile strength, this observation is not surprising. Figure 3f shows the influence of the maximum aggregate size *d_a_* on the normalized shear strenght. The data show a minor decrease in normalized shear strength for increasing maximum aggregate size. Larger aggregates improve the aggregate interlock capacity [143,144], and it is often assumed that using smaller aggregates in small specimens is a conservative approach. For SFRC, however, smaller aggregates result in a more uniform concrete mix with a better bond between the fibers and the concrete.

### 3.2. Comparison to Code Predictions

The experimental shear capacities from the database are then compared to the shear capacities predicted by the code equations and equations proposed in the literature. A difficulty here lies in the definition of the tensile strength of the SFRC, which is based on different experiments depending on local or national practice. As such, it is not possible to build a database containing all values that quantify the tensile behavior of the SFRC, as none of the references report on the outcome of all possible tension tests. As a result, the equations proposed in the literature that were selected for this study depend as much as possible on the concrete compressive strength instead of on the tensile strength.

In a first step, the shear capacity was predicted with 12 sets of equations in total: Sarveghadi et al. [28], Kwak et al. [48], Greenough and Nehdi [50], Khuntia et al. [51], Imam et al. [63], Sharma [52], Mansur et al. [54], Ashour et al. [59]—first equation, Ashour et al. [59]—second set of equations, Arslan et al. [60], Yakoub [64]—first set of equations, and Yakoub [64]—second set of equations. Table 1 contains all expressions. The expression by Greenough and Nehdi [50] uses the reinforcement ratio *ρ* as a percentage instead of as a number. Figure 4 shows the comparison between tested and predicted results, with the statistical properties of *V_utot_*/*V_pred_* in Table 3. Parametric studies for the influence of the different parameters are reported elsewhere [142]. Since not all proposed equations are (explicitly) valid for deep beams, the results for slender beams only are given in Table 4. For all datapoints, the expressions by Kwak et al. [48] result in the smallest coefficient of variation on the ratio of tested to predicted shear capacities and the mean value of tested to predicted shear capacity closest to 1.00, see Table 3. When only the slender beams are considered, the expressions by Arslan et al. [60] result in the smallest coefficient of variation on the tested to predicted shear capacities, combined with an average value of tested to predicted shear capacity close to 1.00 (1.04), see Table 4. In general, the scatter on the tested to predicted shear capacities is high. None of the expressions predicted in the literature is based on a mechanical model that studies the shear-carrying capacity of SFRC based on the mechanisms of shear transfer [18]. The expressions are (semi)-empirical, and thus depend on the database of experiments they were originally derived from. When developing a larger database, as part of this work, the equations do not perform well.

Next, the experimental shear capacities are compared to the code predictions. The code equations that were used for the predictions are the French recommendations [10], the German guideline [11], the *fib* 2010 Model Code [39], and the RILEM recommendations [66]. The predicted shear capacities with the Italian guide [12] are the same as with the *fib* 2010 Model Code [39]; *V_min_* never exceeds the shear capacity of the fiber reinforced concrete. Each of these codes requires the determination of the tensile strength according to experiments described in the respective codes. Since these results are not available in the reported experiments, except for the experiments carried out in the country where the code is valid, the properties had to be calculated. For determination of the tensile strength $f_{cfIk,L2}^{f}$ in the German guideline, the expression from Thomas [145] is used: (57)$$f_{spfc}=0.63\sqrt{f_{cuf}}+0.288\times F\sqrt{f_{cuf}}+0.052\times F$$
To determine $f_{ctR,u}^{f}$, the value of $k_{F}^{f}=0.5$ for shear is used. The value of *C_Rd,c_* = 0.15 is used together with the German guideline, to reflect the German National Annex to the Eurocode, whereas *C_Rd,c_* = 0.18 is used together with the *fib* Model Code provisions and RILEM provisions. For determining *f_Ftuk_* as used in the *fib* Model Code, the value of $f_{ctR,u}^{f}$ from the German code is used. When comparing to the RILEM provisions, it is assumed that *f_Rk,_*_4_ = *f_spfc_* according to Equation (57). For all of the expressions based on the Eurocode shear provisions, the limitation of *ρ* ≤ 2% was removed, so that the heavily reinforced beams from the database could be evaluated as well.

Figure 5 shows the comparison between the tested and predicted shear capacities according to the code equations. For the code equations that are based on the provisions from NEN-EN 1992-1-1:2005 [7], the reduction factor *β* = *a_v_*/*2d* for 0.5*d* ≤ *a_v_* ≤ *2d* is used on the externally applied load but not on the self-weight, to find the sectional shear force at the support *V_utot_*. Table 5 shows the statistical properties of the ratio of the tested to predicted shear capacities. This comparison shows a large scatter on the ratio of experimental to predicted values. For design purposes, the code prediction methods based on the Eurocode shear expression provide reasonable results (with coefficients of variation on the ratio of tested to predicted results of 27–29%). These proposed code equations tend to perform better than the equations proposed in the literature. Full parametric studies based on the tested to predicted shear capacities can be found elsewhere [142].

## 4. Discussion

None of the currently available methods properly describe the behavior of SFRC beams failing in shear, as none of the currently available methods describe the influence of adding steel fibers on the shear-carrying mechanisms: capacity in the compression zone, aggregate interlock, dowel action, residual tension, the contribution of the fibers across the crack, and arching action. This study shows the need for theoretical work that address the different shear-carrying mechanisms in SFRC and its crack kinematics. The large scatter on the ratios of tested to predicted shear capacities found in this study show that the currently available expressions do not describe the shear capacity of SFRC in a satisfactory manner. The code expressions based on the Eurocode are conservative, have smaller scatter as compared to the other expressions, and it seems that these can be used currently for the purpose, as practitioners wait for improved expressions.

An analysis of the ranges of parameters used in the experiments from the literature shows that the majority of tested specimens are small, heavily reinforced for flexure, and tested in three- or four-point bending. Such beams are typical for shear experiments. One may however question how representative such specimens are for actual structural elements. In my opinion, laboratory specimens provide valuable insight into the behavior of SFRC beams failing in shear, but cannot address all open questions. For the implementation of SFRC beams and one-way slabs in buildings and bridges, full-size beams and girders should be designed, and their performance should be evaluated experimentally. Full-size specimens are also required to study the size effect in shear for SFRC.

In earlier work [146], I followed the approach of adding a separate term to quantify the contribution of the steel fibers, in addition to the capacity of the concrete expressed by using the Critical Shear Displacement Theory [147]. This approach is followed by a number of the currently available codes and equations proposed in the literature. However, a further study of the influence of adding steel fibers to the concrete on the shear capacity and the individual shear-carrying mechanisms [18] led me to the conclusion that isolating the contribution of the fibers in a separate, single term is theoretically not correct. The influence of the fibers on all shear-carrying mechanisms should be quantified theoretically, and then evaluated experimentally (for example, with digital image correlation analysis [148,149,150]).

A better understanding of how steel fibers improve the shear resistance of SFRC is important to allow a wider use of SFRC in structural applications. Likewise, a better understanding of the contribution of steel fibers to the shear capacity can result in optimization of cross-sections, a more optimal and economical use of materials, and thus more sustainable designs.

## 5. Summary and Conclusions

One of the barriers for more widespread use of steel fiber reinforced concrete (SFRC) in structural applications, such as beams and girders where part of the stirrups are replaced by fibers, or slabs without stirrups, is the lack of understanding of the shear-carrying behavior. This lack of understanding is reflected by the fact that only a handful of national codes or guidelines contain expressions to quantify the shear capacity of SFRC. This study evaluates the currently available code provisions and equations proposed in the literature for the shear capacity of SFRC elements without stirrups against a database of 488 experimental results from the literature. This study provides an inventory of the current knowledge, identifies the gaps, and proposes a way forward for research on the shear capacity of SFRC elements.

Analyzing the available experimental results from the database resulted in the following conclusions:Most experiments are carried out on small specimens.There is a lack of experiments on SFRC beams with a large depth, which is necessary to evaluate the size effect in shear.Most specimens have a large reinforcement ratio, which is common for shear tests to avoid a flexural failure but does not correspond to actual designs.Experiments on deep and slender beams are available to evaluate the influence of the shear span to depth ratio.The majority of the specimens are cast with normal strength concrete.Most of the fiber volume fractions in the specimens lie between 0.5–1.5% as this range contains practical and workable amounts of fibers and fulfils the aim of partially replacing the mild steel shear reinforcement. The full range of fiber volume fractions in the database is 0.2–4.5%.Historically, different fiber types have been included in experiments. Nowadays, the most commonly used and commercially available fibers are hooked-end fibers. This practice is reflected in the database: 63% of the reported experiments use hooked-end fibers.

Then, parameter studies were carried out based on the available experimental results from the database, which led to the following observations:An analysis of the data shows that the shear stresses should be normalized to the square root of the concrete compressive strength, as this ratio shows a smaller relation to the concrete compressive strength than the cube root of the concrete compressive strength.The normalized shear strength increases as the reinforcement ratio increases, which can be explained by the larger dowel action for larger amounts of reinforcement.The data show a small decrease for the normalized shear strength as the effective depth increases. Not enough experimental results on large SFRC beams are available to study the size effect in shear in SFRC.The influence of the shear span to depth ratio on the normalized shear strength is similar in SFRC as in reinforced concrete. The higher shear strength for small values of the shear span to depth ratio is the result of arching action.The normalized shear strength increases as the fiber volume fraction increases. The normalized shear strength increases as the fiber factor increases. These observations are expected, since the contribution of the fibers improves the shear capacity. There is less scatter on the influence of the fiber factor than on the influence of the fiber volume fraction, which justifies the use of the fiber factor in expressions and code equations.The normalized shear strength decreases as the maximum aggregate size increases. This observation in contrary to what happens in reinforced concrete, where larger aggregates improve the aggregate interlock capacity and thus the shear capacity. In SFRC, smaller aggregates result in a more uniform mix, and a better bond between the concrete matrix and the steel fibers, which enhances the shear capacity.

For the comparison between the experimental shear capacities and the capacities predicted by the currently available codes and equations proposed in the literature, the following conclusions result:National codes and guidelines are based on specific methods for determining the tensile strength of the SFRC, and these methods differ internationally. As such, none of the experiments available in the literature report on all values of the tensile strength that are required for determining the tensile strength in the various expressions.The ratio of tested to predicted shear capacities shows large scatter. When all experiments are considered, the expression by Kwak et al. results in the best performance. When only slender beams are considered, the expression by Arslan et al. results in the best performance.The code equations based on the Eurocode shear expressions have a coefficient of variation between 27% and 29% and a slightly conservative value of the average ratio of the tested to predicted shear capacity. As such, these equations can be used until better proposals are available.

The analysis in this work shows the need for a better understanding of the shear capacity of SFRC. An analysis of the influence of the steel fibers on all shear-carrying mechanisms seems necessary. A better understanding of the shear-carrying mechanisms is necessary to allow a more widespread use of SFRC in structural elements, and an optimization of designs.

## Figures and Tables

**Figure 1 materials-12-00917-f001:**
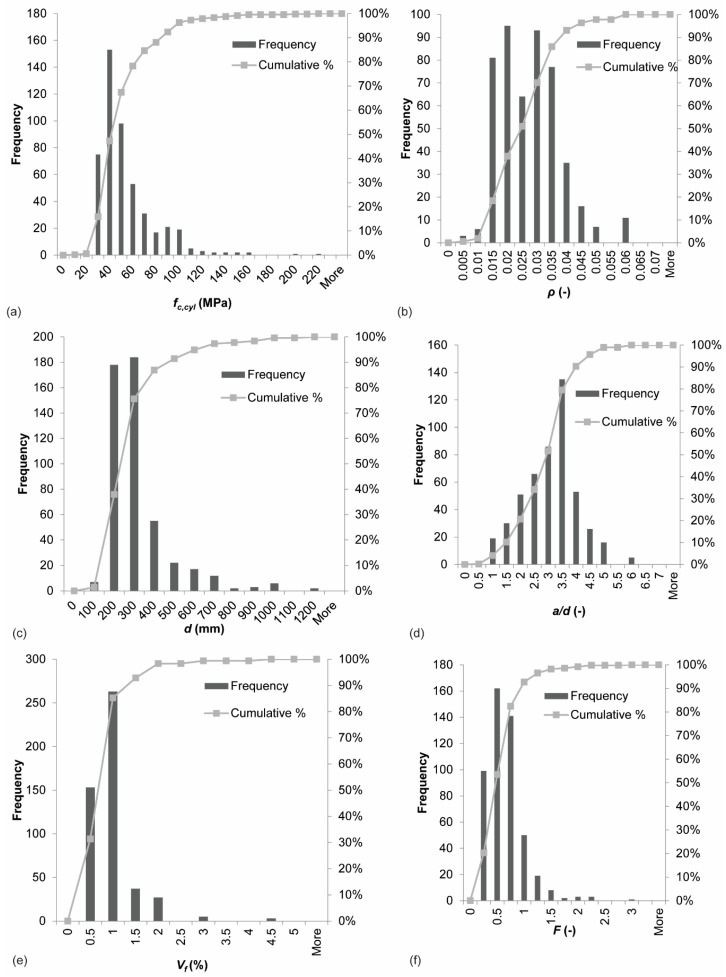
Distribution of parameters in database: (**a**) concrete compressive strength *f_c,cyl_*; (**b**) reinforcement ratio *ρ*; (**c**) effective depth *d*; (**d**) shear span to depth ratio *a*/*d*; (**e**) fiber volume fraction *V_f_*; and (**f**) fiber factor *F*.

**Figure 2 materials-12-00917-f002:**
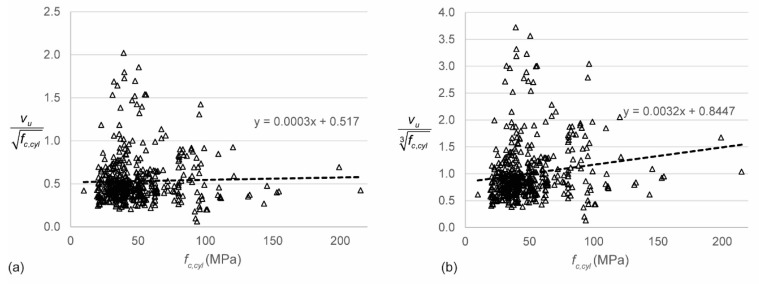
Normalized shear stresses to the concrete compressive strength: (**a**) normalized to the square root; (**b**) normalized to the cube root.

**Figure 3 materials-12-00917-f003:**
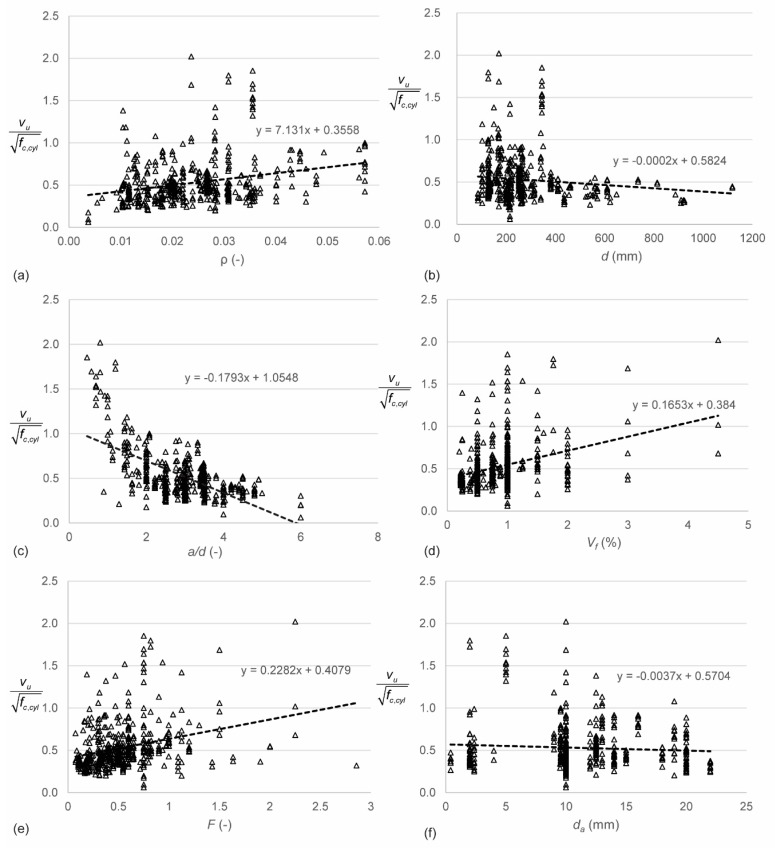
Parameter studies based on normalized shear stresses for all entries in database, influence of (**a**) longitudinal reinforcement ratio *ρ*; (**b**) effective depth *d*; (**c**) shear span to depth ratio *a*/*d*; (**d**) fiber volume fraction *V_f_*; (**e**) fiber factor *F*; and (**f**) maximum aggregate size *d_a_*.

**Figure 4 materials-12-00917-f004:**
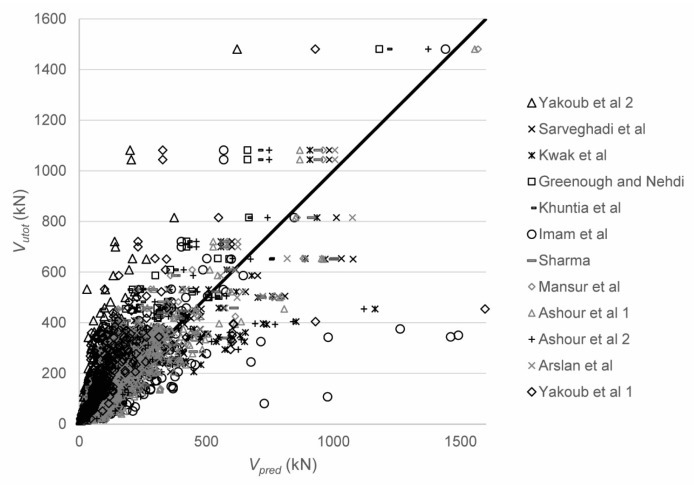
Comparison between experimental and predicted shear capacities for 12 methods from the literature.

**Figure 5 materials-12-00917-f005:**
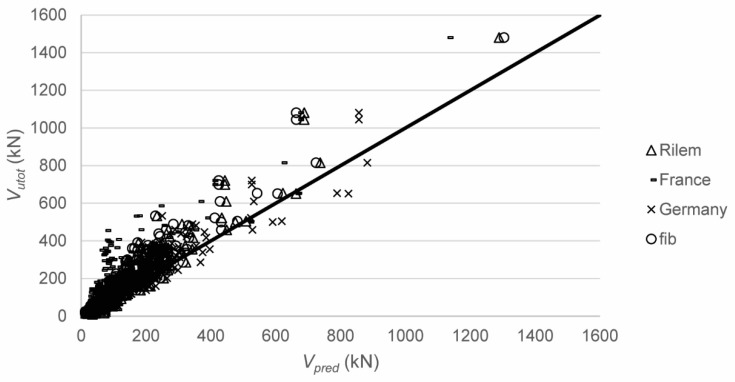
Comparison between tested and predicted shear capacities with the code formulas.

**Table 1 materials-12-00917-t001:** Shear prediction equations from literature and available codes.

Authors	Reference	Expression	
Sarveghadi et al.	[28]	$V_{u}=\left[\rho +\frac{\rho }{v_{b}}+\frac{1}{\frac{a}{d}}\left(\frac{\rho f_{t}{}^{\prime }\left(\rho +2\right)\left(f_{t}{}^{\prime }\frac{a}{d}-\frac{3}{v_{b}}\right)}{\frac{a}{d}}+f_{t}{}^{\prime }\right)+v_{b}\right]b_{w}d$	(2)
		$f_{t}{}^{\prime }=0.79\sqrt{f_{c}{}^{\prime }}$	(3)
		$v_{b}=0.41\tau F$ with *τ* = 4.15 MPa	(4)
Kwak et al.	[48]	$V_{u}=\left[3.7ef_{spfc}^{2/3}{\left(\rho \frac{d}{a}\right)}^{1/3}+0.8v_{b}\right]b_{w}d$	(5)
		$f_{spfc}=\frac{f_{cuf}}{\left(20-\sqrt{F}\right)}+0.7+1.0\sqrt{F}$ in MPa	(6)
		$e=\left\{ \begin{array}{l} 1\;\mathrm{for}\;\frac{a}{d}>3.4\\ 3.4\frac{d}{a}\;\mathrm{for}\;\frac{a}{d}\leq 3.4 \end{array}\right.$	(7)
Greenough and Nehdi	[50]	$V_{u}=\left[0.35\left(1+\sqrt{\frac{400}{d}}\right){\left(f_{c}{}^{\prime }\right)}^{0.18}{\left(\left(1+F\right)\rho \frac{d}{a}\right)}^{0.4}+0.9\eta_{o}\tau F\right]b_{w}d$	(8)
Kuntia et al.	[51]	$V_{u}=\left[\left(0.167+0.25F\right)\sqrt{f_{c}{}^{\prime }}\right]b_{w}d$	(9)
Sharma	[52]	$V_{u}=\left(\frac{2}{3}\times 0.8\sqrt{f_{c}{}^{\prime }}{\left(\frac{d}{a}\right)}^{0.25}\right)b_{w}d$	(10)
Mansur et al.	[54]	$V_{u}=V_{c}+\sigma_{tu}b_{w}d$	(11)
		$V_{c}=\left(0.16\sqrt{f_{c}{}^{\prime }}+17.2\frac{\rho Vd}{M}\right)b_{w}d\leq 0.29\sqrt{f_{c}{}^{\prime }}b_{w}d$	(12)
		$\sigma_{tu}=3.2\eta_{o}\eta_{l}F\tau \;\mathrm{with}\;\tau =2.58\;\mathrm{MPa}$	(13)
		$\eta_{l}=1-\frac{\tanh\left(\beta \frac{l_{f}}{2}\right)}{\beta \frac{l_{f}}{2}}$	(14)
		$\beta =\sqrt{\frac{2\pi G_{m}}{E_{f}A_{f}\ln\left(\frac{S}{r_{f}}\right)}}$	(15)
		$S=25\sqrt{\frac{d_{f}}{V_{f}l_{f}}}$	(16)
Ashour et al.	[59]	$V_{u}=\left[\left(0.7\sqrt{f_{c}{}^{\prime }}+7F\right)\frac{d}{a}+17.2\rho \frac{d}{a}\right]b_{w}d$	(17)
		$V_{u}=\left[\left(2.11\sqrt[{3}]{f_{c}{}^{\prime }}+7F\right){\left(\rho \frac{d}{a}\right)}^{0.333}\right]b_{w}d\;\mathrm{for}\;\frac{a}{d}\geq 2.5$	(18)
		$V_{u}=\left[\left(\left(2.11\sqrt[{3}]{f_{c}{}^{\prime }}+7F\right){\left(\rho \frac{d}{a}\right)}^{0.333}\right)\frac{2.5}{\frac{a}{d}}+v_{b}\left(2.5-\frac{a}{d}\right)\right]b_{w}d\;\mathrm{for}\;\frac{a}{d}<2.5$	(19)
Arslan et al.	[60]	$V_{u}=\left[\left(0.2{\left(f_{c}{}^{\prime }\right)}^{2/3}\frac{c}{d}+\sqrt{\rho \left(1+4F\right)f_{c}{}^{\prime }}\right)\sqrt[{3}]{\frac{3}{\frac{a}{d}}}\right]b_{w}d$	(20)
		${\left(\frac{c}{d}\right)}^{2}+\left(\frac{600\rho }{f_{c}{}^{\prime }}\right)\left(\frac{c}{d}\right)-\frac{600\rho }{f_{c}{}^{\prime }}=0$	(21)
Imam et al.	[63]	$V_{u}=\left[0.6\psi \sqrt[{3}]{\omega }\left({\left(f_{c}{}^{\prime }\right)}^{0.44}+275\sqrt{\frac{\omega }{{\left(\frac{a}{d}\right)}^{5}}}\right)\right]b_{w}d$	(22)
		$\psi =\frac{1+\sqrt{\frac{5.08}{d_{a}}}}{\sqrt{1+\frac{d}{25d_{a}}}}$	(23)
		ω=ρ(1+4F)	(24)
Yakoub	[64]	$V_{u}=\left[0.83\xi \sqrt[{3}]{\rho }\left(\sqrt{f_{c}{}^{\prime }}+249.28\sqrt{\frac{\rho }{{\left(\frac{a}{d}\right)}^{5}}}+0.405\frac{l_{f}}{d_{f}}V_{f}R_{g}\frac{d}{a}\sqrt{f_{c}{}^{\prime }}\right)\right]b_{w}d\;\mathrm{for}\;\frac{a}{d}\leq 2.5$	(25)
		$V_{u}=\left[0.83\xi \sqrt[{3}]{\rho }\left(\sqrt{f_{c}{}^{\prime }}+249.28\sqrt{\frac{\rho }{{\left(\frac{a}{d}\right)}^{5}}}+0.162\frac{l_{f}}{d_{f}}V_{f}R_{g}\sqrt{f_{c}{}^{\prime }}\right)\right]b_{w}d\;\mathrm{for}\;\frac{a}{d}\geq 2.5$	(26)
		$\xi =\frac{1}{\sqrt{1+\frac{d}{25d_{a}}}}$	(27)
		$V_{u}=2.5\left(\frac{0.40}{1+1500\varepsilon_{x}}\times \frac{1300}{1000+s_{xe}}\right)\sqrt{f_{c}{}^{\prime }}\left(1+0.7\frac{l_{f}}{d_{f}}V_{f}R_{g}\right)\frac{d}{a}b_{w}d_{v}\;\mathrm{for}\;\frac{a}{d}\leq 2.5$	(28)
		$V_{u}=\left(\frac{0.40}{1+1500\varepsilon_{x}}\times \frac{1300}{1000+s_{xe}}\right)\sqrt{f_{c}{}^{\prime }}\left(1+0.7\frac{l_{f}}{d_{f}}V_{f}R_{g}\right)b_{w}d_{v}\;\mathrm{for}\;\frac{a}{d}\geq 2.5$	(29)
		$d_{v}=\max\left(0.9d,0.72h\right)$	(30)
		$\varepsilon_{x}=\frac{\frac{M}{d_{v}}+V}{2E_{s}A_{s}}$	(31)
		$s_{xe}=\frac{35s_{x}}{16+d_{a}}\geq 0.85s_{x}\;\mathrm{and}\;s_{x}\approx d_{v}$	(32)
Association Française de Génie Civil	[10]	$V_{Rd}=V_{Rd,c}+V_{Rd,f}$	(33)
		$V_{Rd,c}=\frac{0.21}{\gamma_{cf}\gamma_{E}}f_{ck}^{1/2}b_{w}d$	(34)
		$V_{Rd,f}=\frac{A_{vf}\sigma_{Rd,f}}{\tan\theta }$	(35)
		$\sigma_{Rd,f}=\left\{ \begin{array}{l} \frac{1}{K\gamma_{cf}}\frac{1}{w_{\lim}}\int_{0}^{w_{\lim}}\sigma_{f}\left(w\right)dw\;\mathrm{for}\;\mathrm{strain}\;\mathrm{softening}\;\mathrm{or}\;\mathrm{low}\;\mathrm{strain}\;\mathrm{hardening}\\ \frac{1}{K\gamma_{cf}}\frac{1}{\varepsilon_{\lim}-\varepsilon_{el}}\int_{\varepsilon_{el}}^{\varepsilon_{\lim}}\sigma_{f}\left(\varepsilon \right)d\varepsilon \;\mathrm{for}\;\mathrm{high}\;\mathrm{strain}\;\mathrm{hardening} \end{array}\right.$	(36)
		$w_{\lim}=\max\left(w_{u},w_{\max}\right)$	(37)
		$\varepsilon_{\lim}=\max\left(\varepsilon_{u},\varepsilon_{\max}\right)$	(38)
		$A_{vf}=b_{w}z$	(39)
DAfStB	[11]	$V_{Rd,c}^{f}=V_{Rd,c}+V_{Rd,cf}$	(40)
		$V_{Rd,c}=\frac{C_{Rd,c}}{\gamma_{c}}k{\left(100\rho f_{ck}\right)}^{1/3}b_{w}d>V_{Rd,c,\min}$	(41)
		$V_{Rd,cf}=\frac{\alpha_{c}^{f}f_{ctR,u}^{f}b_{w}h}{\gamma_{ct}^{f}}$	(42)
		$f_{ctR,u}^{f}=k_{F}^{f}k_{G}^{f}0.37f_{cfIk,L2}^{f}$	(43)
		$k_{G}^{f}=1.0+0.5A_{ct}^{f}\leq 1.7$	(44)
		$A_{ct}^{f}=b_{w}\times \min\left(d,1.5m\right)$	(45)
		$k=1+\sqrt{\frac{200mm}{d}}$	(46)
RILEM	[66]	$V_{Rd}=V_{cd}+V_{fd}$	(47)
		$V_{cd}=0.12k{\left(100\rho f_{ck}\right)}^{\frac{1}{3}}b_{w}d$	(48)
		$V_{fd}=0.7k_{f}k\tau_{fd}b_{w}d$	(49)
		$k_{f}=1+n\left(\frac{h_{f}}{b_{w}}\right)\left(\frac{h_{f}}{d}\right)\leq 1.5$	(50)
		$n=\frac{b_{f}-b_{w}}{h_{f}}\leq 3\;\mathrm{and}\;n\leq \frac{3b_{w}}{h_{f}}$	(51)
		$\tau_{fd}=0.12f_{Rk,4}$	(52)
*fib*	[39]	$V_{Rd}=V_{Rd,f}=\frac{C_{Rd,c}}{\gamma_{c}}k{\left(100\rho \left(1+7.5\frac{f_{Ftuk}}{f_{ctk}}\right)f_{ck}\right)}^{1/3}b_{w}d$	(53)
		$f_{ctk}=\left\{ \begin{array}{l} 0.3{\left(f_{ck}\right)}^{2/3}\;\mathrm{for}\;\mathrm{concrete}\;\mathrm{grades}\;\leq \;C50\\ 2.12\ln\left(1+0.1\left(f_{ck}+8MPa\right)\right)\mathrm{for}\;\mathrm{concrete}\;\mathrm{grades}\;>\;C50 \end{array}\right.$	(54)
CNR-DT	[12]	$V_{Rd}=V_{Rd,f}\geq V_{min}$	(55)
		$V_{min}=0.035k^{3/2}f_{ck}^{1/2}b_{w}d$	(56)

**Table 2 materials-12-00917-t002:** Ranges of parameters in database.

Parameter	Min	Max
*b*_w_ (mm)	50	610
*h* (mm)	100	1220
*d* (mm)	85	1118
*l_span_* (mm)	204	7823
*a* (mm)	102	3912
*a_v_* (mm)	52	3747
*ρ* (%)	0.37%	5.72%
*f_y_* (MPa)	276	900
*a*/*d* (-)	0.46	6
*a_v_*/*d* (-)	0.20	5.95
*d_a_* (mm)	0.4	22
*f_c,cyl_* (MPa)	9.8	215
*V_f_* (%)	0.2	4.5
*l_f_*/*d_f_* (-)	25	191
*f_tenf_* (MPa)	260	4913
*F* (-)	0.075	2.858

**Table 3 materials-12-00917-t003:** Statistical properties of *V_utot_*/*V_pred_* for all 488 datapoints, with AVG = average of *V_utot_/V_pred_*, STD = standard deviation on *V_utot_/V_pred_*, and COV = coefficient of variation of *V_utot_/V_pred_*.

Model	AVG	STD	COV	Min	Max
Sarveghadi et al. [28]	1.03	0.29	28%	0.23	2.49
Kwak et al. [48]	1.01	0.28	27%	0.27	2.39
Greenough and Nehdi [50]	1.34	0.48	36%	0.31	3.11
Khuntia et al. [51]	1.81	0.85	47%	0.18	6.53
Imam et al. [63]	0.97	0.36	37%	0.06	2.51
Sharma [52]	1.24	0.49	39%	0.18	3.59
Mansur et al. [54]	1.30	0.60	46%	0.15	3.85
Ashour et al. [59] 1	1.08	0.38	35%	0.24	3.14
Ashour et al. [59] 2	1.29	0.37	29%	0.31	3.22
Arslan et al. [60]	1.17	0.37	31%	0.43	3.24
Yakoub [64] 1	1.90	0.76	40%	0.28	7.50
Yakoub [64] 2	2.97	1.37	46%	0.51	17.48

**Table 4 materials-12-00917-t004:** Statistical properties of *V_test_*/*V_pred_* for 352 datapoints with *a*/*d* ≥ 2.5, with AVG = average of *V_utot_/V_pred_*, STD = standard deviation on *V_utot_/V_pred_*, and COV = coefficient of variation of *V_utot_/V_pred_*.

Model	AVG	STD	COV	Min	Max
Sarveghadi et al. [28]	1.02	0.29	28%	0.23	2.20
Kwak et al. [48]	1.06	0.28	26%	0.27	2.39
Greenough and Nehdi [50]	1.20	0.37	30%	0.31	3.11
Khuntia et al. [51]	1.53	0.48	31%	0.18	4.03
Imam et al. [63]	1.07	0.31	29%	0.32	2.51
Sharma [52]	1.11	0.33	30%	0.18	2.28
Mansur et al. [54]	1.12	0.42	38%	0.15	3.57
Ashour et al. [59] 1	1.15	0.40	35%	0.24	3.14
Ashour et al. [59] 2	1.35	0.35	26%	0.47	3.22
Arslan et al. [60]	1.04	0.24	23%	0.43	1.97
Yakoub [64] 1	2.03	0.80	39%	0.62	7.50
Yakoub [64] 2	2.83	1.37	49%	0.61	17.48

**Table 5 materials-12-00917-t005:** Statistical properties of *V_utot_*/*V_pred_* for all 488 datapoints, with AVG = average of *V_utot_/V_pred_*, STD = standard deviation on *V_utot_/V_pred_*, and COV = coefficient of variation of *V_utot_/V_pred_*.

Model	AVG	STD	COV	Min	Max
French code [10]	1.85	0.88	48%	0.22	5.95
German code [11]	1.12	0.31	27%	0.21	2.13
*fib* [39]	1.24	0.36	29%	0.30	2.33
RILEM [66]	1.16	0.33	29%	0.23	2.28

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
