# Peer review of "Database of Shear Experiments on Steel Fiber Reinforced Concrete Beams without Stirrups"

_materials, 2019, doi:10.3390/ma12060917_

Reviewer 1 Report

The paper is aimed at reporting a huge database and explores the soundness of existing empirical equations and code provisions for the prediction of the shear capacity of SFRC beams (reinforced without conventional rebars) under shear failure. The work contains outstanding materials and very complete (it is almost evident that the author did a huge effort for collecting the manuscript contents). It is recommended its publishing.

In the Reviewer's opinion the paper needs few minor improvements before the approval.

The following recommendations/clarifications should be considered:

·       Title: I would suggest to modify a bit it since in the current version the title only reports “the database of shear experiments of SFRC”. The Reviewer also suggests to remark, in the title, that the work also deals with exploring the soundness and capabilities of existing empirical equations and code provisions to predict SFRC shear force failures.

·       The Reviewer also suggests to include the formula suggested by the Italian Guidelines CNR-DT 204/2006 (2006) for FRC in Table 1.

·       One of the parameters considered in the database was also the fiber type (e.g., hooked-end, twisted, crimped, straight, etc.). This should be accounted in Table 2.

·       Figure 1 should be also completed by showing the frequency of experimental data which considered different fiber types: hooked-end fibers (I guess the majority of the case) and/or other ones.

·       The author, between lines 425 and 430, wrote that most of the fiber volume fractions in tested specimens lie between 0.5% - 1.5%, as this range contains practical and workable amounts of fibers. This is only partially true. The reason of these low amounts of fibers is also motivated by the fact that in the practice SFs are always used in partially substitution of classical rebars. They are almost never used alone, for structural purposes.

·       Please add another item comment, when the conclusion are drawn out (i.e., after lines 416-417), by analyzing the available experimental results and referring on the effect of the fiber type analyzed.

·       Future developments which will follow to this research paper are only poorly outlined. It should be reported at the end of the concluding section.

Author Response

Reviewer 1

The paper is aimed at reporting a huge database and explores the soundness of existing empirical equations and code provisions for the prediction of the shear capacity of SFRC beams (reinforced without conventional rebars) under shear failure. The work contains outstanding materials and very complete (it is almost evident that the author did a huge effort for collecting the manuscript contents). It is recommended its publishing.

Thank you for your kind comments.

In the Reviewer's opinion the paper needs few minor improvements before the approval.

Thank you – I have improved the manuscript with your suggestions and the suggestions of the other reviewers.

The following recommendations/clarifications should be considered:

·       Title: I would suggest to modify a bit it since in the current version the title only reports “the database of shear experiments of SFRC”. The Reviewer also suggests to remark, in the title, that the work also deals with exploring the soundness and capabilities of existing empirical equations and code provisions to predict SFRC shear force failures.

My alternative title would be: “Shear capacity of steel fiber reinforced concrete beams without stirrups: database and evaluation of prediction methods” – but this title is more than 12 words. I’ve proposed this title to the editor to let her make a call of judgement here.

·       The Reviewer also suggests to include the formula suggested by the Italian Guidelines CNR-DT 204/2006 (2006) for FRC in Table 1.

Thank you for the suggestion – I was actually not aware of the Italian guide! Now I also see where the fib Model Code expressions come from. I’ve included the expression in Table 1 and included it in the text at various points:

In the introduction:

The most noteworthy national codes and guidelines with shear provisions for SFRC are the French recommendations [8-10], the German guideline [11], and the Italian guide [12].

Above Table 1:

The expressions from the German guideline [11] and RILEM [66] are based on the Eurocode EN 1992-1-1:2004 [7] equations, by adding a term to represent the contribution of the steel fibers. The expressions from the fib Model Code [40] are based on EN 1992-1-1:2004 [7], but incorporate the effect of the fibers into the original expression. The Italian guide [12] uses the same expressions as the fib Model Code  [40], and includes a lower bound for the shear capacity Vmin. In the German National Annex of the Eurocode 2, CRd,c = 0.15, and this value is used in Eq. as well. The following factors are used: γc = 1.5,  = 1.25,  = 0.85 to account for long-term effects, and  = 0.5 for shear. For cross-sections subjected to axial loads, the contribution of the steel fibers cannot be taken into account, as more experimental results are necessary to derive suitable expressions [24]. In the Italian guideline [12], the influence of axial loads is considered in the same way as in EN 1992-1-1:2004 [7]. Since this work deals with elements without axial loads, the formulas have been simplified accordingly. The expressions from the German guideline [11], RILEM [66], the fib Model Code [40], and the Italian guide [12] are valid for ρ ≤ 2%. For the fib Model Code expressions, CRd,c = 0.18 and γc = 1.5. All notations used in Table 1 are explained in the “List of notations”.    

Under Table 4:

Next, the experimental shear capacities are compared to the code predictions. The code equations that were used for the predictions are the French recommendations [10], the German guideline [11], the fib 2010 Model Code [40], and the RILEM recommendations [66]. The predicted shear capacities with the Italian guide [12] are the same as with the fib 2010 Model Code [40]; Vmin never exceeds the shear capacity of the fiber reinforced concrete.

·       One of the parameters considered in the database was also the fiber type (e.g., hooked-end, twisted, crimped, straight, etc.). This should be accounted in Table 2.

I added the following discussion in the section above Table 2:

The fiber types that occur in the database are: hooked, crimped, straight smooth, mixed (hooked + straight), fibers with a flat end, flat fibers, round fibers, mill-cut fibers, fibers of straight mild steel, brass-coated high strength steel fibers, chopped fibers with butt ends, recycled fibers, and corrugated fibers. The most frequently used fibers in the database are hooked (63% of all gathered experiments), crimped (22% of experiments), and straight smooth (3%). 

·       Figure 1 should be also completed by showing the frequency of experimental data which considered different fiber types: hooked-end fibers (I guess the majority of the case) and/or other ones.

Since fiber type is not a quantitative value like the other parameters in Figure 1, I have opted for adding a paragraph discussing the fiber type instead.

The fiber types that occur in the database are: hooked, crimped, straight smooth, mixed (hooked + straight), fibers with a flat end, flat fibers, round fibers, mill-cut fibers, fibers of straight mild steel, brass-coated high strength steel fibers, chopped fibers with butt ends, recycled fibers, and corrugated fibers. The most frequently used fibers in the database are hooked (63% of all gathered experiments), crimped (22% of experiments), and straight smooth (3%). 

As you expected, the majority of the fibers (63% of the experiments in my database) are hooked-end fibers.

·       The author, between lines 425 and 430, wrote that most of the fiber volume fractions in tested specimens lie between 0.5% - 1.5%, as this range contains practical and workable amounts of fibers. This is only partially true. The reason of these low amounts of fibers is also motivated by the fact that in the practice SFs are always used in partially substitution of classical rebars. They are almost never used alone, for structural purposes.

Good point. I’ve added this in the summary and conclusions as follows:

·         Most of the fiber volume fractions in the specimens lie between 0.5% - 1.5% as this range contains practical and workable amounts of fibers and fulfils the aim of partially replacing the mild steel reinforcement. The full range of fiber volume fractions in the database is 0.2% - 4.5%.

and in the discussion of the parameter ranges in the database:

The histogram of the fiber volume fraction Vf, Figure 1e, shows crowding in the range of 0.5% - 1.5%. This observation is not surprising, as these fractions are practical values: these fractions result in workable mixes, and serve the purpose of partially (not fully) replacing the mild steel reinforcement.

·       Please add another item comment, when the conclusion are drawn out (i.e., after lines 416-417), by analyzing the available experimental results and referring on the effect of the fiber type analyzed.

I added the following bullet point:

·         Historically, different fiber types have been included in experiments. Nowadays, the most commonly used and commercially available fibers are hooked-end fibers. This practice is reflected in the database: 63% of the reported experiments use hooked-end fibers.

·       Future developments which will follow to this research paper are only poorly outlined. It should be reported at the end of the concluding section.

At the end of the concluding section, I added a short outline of my future work:

To address this challenge, future work will include shear experiments of SFRC beams with DIC measurements to study the crack kinematics and contributions to the shear capacity of the different shear-carrying mechanisms, combined with theoretical work.

Reviewer 2 Report

This would be probably the best manuscript that we have ever reviewed for Materials, if the discussions were further developed. The subject is interesting and belongs to a research field where investigation is still much needed. Gaps in existing knowledge are identified by the author, stressing the relevance of the presents work. It encompasses useful information collated from an impressive literature review.

Nevertheless, some minor suggestions are provided hereunder.

Ln 20 coefficients of variation on the ratio tested/predicted

Ln 27 “may be improved” instead of “are improved”

Ln 96 Reference for the source of eq. (1) seems missing

Ln 102 and throughout “Similarly, Sharma’s proposal [51]” instead of “Similarly, Sharma [51]’s proposal” seems preferable

Ln 269 “concrete?” or “concrete compressive strength?” ?

Ln 270 “I analyzed the normalized shear stress” or “The normalized shear stress was analysed”

It is arguable if in fig. 3c the relationship is linear or power (negative)

Ln 334 “parameter studies” or “parametric studies”?

Ln 343 “derived for” or “derived from”?

Why were the found laws evaluated only in terms of prediction c.v. and not also by their mean accuracy?

German code’s model seems to perform quite well!

Ln 392 “practitioners wait” instead of “the profession waits” is suggested

Ln 393 “I followed the approach of adding a separate term” or “the approach of adding a separate term… was followed”

Ln 396 “brought me” or “led”

Author Response

Reviewer 2

This would be probably the best manuscript that we have ever reviewed for Materials, if the discussions were further developed. The subject is interesting and belongs to a research field where investigation is still much needed. Gaps in existing knowledge are identified by the author, stressing the relevance of the presents work. It encompasses useful information collated from an impressive literature review.

Thank you for your kind comments.

I have extended the discussion with the following paragraph:

An analysis of the ranges of parameters used in the experiments from the literature shows that the majority of tested specimens are small, heavily reinforced for flexure, and tested in three- or four-point bending. Such beams are typical for shear experiments. One may however question how representative such specimens are for actual structural elements. In my opinion, laboratory specimens provide valuable insight in the behavior of SFRC beams failing in shear, but cannot address all open questions. For the implementation of SFRC beams in buildings and bridges, full-size beams and girders should be designed, and their performance should be evaluated experimentally. Full-size specimens are also required to study the size effect in shear for SFRC.

Nevertheless, some minor suggestions are provided hereunder.

Thank you for your comments. I have improved the manuscript with your comments and the comments of the other reviewers.

Ln 20 coefficients of variation on the ratio tested/predicted

Changed to:

with coefficients of variation on the ratio tested/predicted shear capacities of 27% - 29%

Ln 27 “may be improved” instead of “are improved”

changed

Ln 96 Reference for the source of eq. (1) seems missing

I added the reference to Narayanan and Kareem-Palanjian from 1984.

Ln 102 and throughout “Similarly, Sharma’s proposal [51]” instead of “Similarly, Sharma [51]’s proposal” seems preferable

You are right – I have changed this accordingly.

Ln 269 “concrete?” or “concrete compressive strength?” ?

Changed to: concrete cylinder compressive strength

Ln 270 “I analyzed the normalized shear stress” or “The normalized shear stress was analysed”

I’m trying to reduce my use of passive voice, so I’ll keep the active voice here.

It is arguable if in fig. 3c the relationship is linear or power (negative)

It’s hard to tell – looking at it again, it seems bilinear, with a cut-off point for a/d somewhere between 2 and 3.5. I added a small discussion there:

 Figure 3c shows the influence of the shear span to depth ratio in terms of a/d. Note that the linear relation plotted on the graph is presented for consistency with the other figures, but does not accurately present the relation between a/d and the normalized shear strength. The results show that, just as for reinforced concrete beams, the shear capacity for specimens with a/d ≤ 2.5 increases for a decrease in a/d.

Ln 334 “parameter studies” or “parametric studies”?

parametric studies is indeed better, I’ve changed this.

Ln 343 “derived for” or “derived from”?

Derived from is indeed better.

Why were the found laws evaluated only in terms of prediction c.v. and not also by their mean accuracy?

I added this as follows:

For all datapoints, the expressions by Kwak et al. [49] result in the smallest coefficient of variation on the ratio of tested to predicted shear capacities and the mean value of tested to predicted shear capacity closest to 1.00, see Table 4. When only the slender beams are considered, the expressions by Arslan et al. [61] result in the smallest coefficient of variation on the tested to predicted shear capacities, combined with an average value of tested to predicted shear capacity close to 1.00 (1.04), see Table 6

German code’s model seems to perform quite well!

Yes.

Ln 392 “practitioners wait” instead of “the profession waits” is suggested

I changed this.

Ln 393 “I followed the approach of adding a separate term” or “the approach of adding a separate term… was followed”

I’d like to keep active voice here.

Ln 396 “brought me” or “led”

Changed to “led”

Reviewer 3 Report

Dear authors,

The paper is very interesting and documented.

The authors highlight the existing empirical equations and code provisions utilizate for the prediction of the shear capacity of SFRC beams failing in shear as well as a database of 487 experiments reported of the scientific researcher in the 152 studied references.

However, I have to comment and suggest at the same time that the Discussion section should have been much broader and the section Conclusions should be shorter.  

Consequently, the paper meets the publishing standards as Literature Review and is therefore useful for many researchers.

Author Response

Reviewer 3

Dear authors,

The paper is very interesting and documented.

Thank you for your kind comments.

I have prepared a revised version of the manuscript, taking into account your comments and the comments of the other reviewers.

The authors highlight the existing empirical equations and code provisions utilizate for the prediction of the shear capacity of SFRC beams failing in shear as well as a database of 487 experiments reported of the scientific researcher in the 152 studied references.

However, I have to comment and suggest at the same time that the Discussion section should have been much broader and the section Conclusions should be shorter.  

I have added a paragraph to the discussion:

An analysis of the ranges of parameters used in the experiments from the literature shows that the majority of tested specimens are small, heavily reinforced for flexure, and tested in three- or four-point bending. Such beams are typical for shear experiments. One may however question how representative such specimens are for actual structural elements. In my opinion, laboratory specimens provide valuable insight in the behavior of SFRC beams failing in shear, but cannot address all open questions. For the implementation of SFRC beams in buildings and bridges, full-size beams and girders should be designed, and their performance should be evaluated experimentally. Full-size specimens are also required to study the size effect in shear for SFRC.

Please note that the Conclusions are a “Summary and conclusions”, and as such contain a summary of the work carried out as well as the main conclusions of the work.

Consequently, the paper meets the publishing standards as Literature Review and is therefore useful for many researchers.

Thank you.